# Survey on Endoparasites of Dairy Goats in North-Eastern Italy Using a Farm-Tailored Monitoring Approach

**DOI:** 10.3390/vetsci8050069

**Published:** 2021-04-22

**Authors:** Anna Maurizio, Laura Stancampiano, Cinzia Tessarin, Alice Pertile, Giulia Pedrini, Ceren Asti, Waktole Terfa, Antonio Frangipane di Regalbono, Rudi Cassini

**Affiliations:** 1Department of Animal Medicine, Production and Health, University of Padova, Viale dell’Università, 16-35020 Legnaro, Italy; cinzia.tessarin@unipd.it (C.T.); antonio.frangipane@unipd.it (A.F.d.R.); rudi.cassini@unipd.it (R.C.); 2Department of Veterinary Medical Sciences, University of Bologna, Via Tolara di Sopra, 50-40064 Ozzano dell’Emilia, Italy; laura.stancampiano@unibo.it; 3SIVAR, 36030 Valli del Pasubio, Italy; alice.pertile@libero.it; 4SIVAR, 35100 Padova, Italy; giulia.pedrini.vet@gmail.com; 5Department of Parasitology, Faculty of Veterinary Medicine, Ankara University, Ankara 06110, Turkey; Ceren.Asti@ankara.edu.tr; 6Department of Veterinary Science, Mamo Mezemir Campus, School of Veterinary Medicine, University of Ambo, Guder, Ethiopia; waqtee@gmail.com; 7College of Veterinary Medicine and Agriculture, Addis Ababa University, Bishoftu P.O. Box 34, Ethiopia

**Keywords:** dairy goats, endoparasites, Italy, sample size, aggregation

## Abstract

With the spread of anthelmintic resistance (AR), endoparasite monitoring consolidates its role for a more sustainable targeting of treatments. A survey on endoparasites in dairy goat farms of north-eastern Italy was conducted to test a monitoring approach based on a farm-tailored sample size. Farm management and parasites control practices were investigated in 20 farms through a questionnaire survey. Further, fecal samples were collected (November 2018–September 2019) from 264 animals from 13 farms and were analyzed individually with a modified McMaster method and subsequently pooled to perform a coproculture. Coccidia (78.4%), gastrointestinal strongyles (37.9%), *Strongyloides* (28.4%), *Skrjabinema* (18.9%), *Trichuris* (8.0%) and *Nematodirus*/*Marshallagia* (0.4%) were identified. Abundances were higher for coccidia and gastrointestinal strongyles. *Haemonchus* (71%) was the dominant gastrointestinal nematode. Pasture and age class resulted in the main risk factors at the multivariable analysis through a negative binomial regression model. Results from farm monitoring indicate that our approach can be a cost-effective decision tool to target treatments more effectively, but farmers need to be educated about the importance of parasitological testing, which is currently scarcely implemented, against the risk of AR.

## 1. Introduction

Goat farming plays a fundamental role in economic, environmental and cultural perspectives in both developed and developing countries [1]. Among the diseases affecting goats, endoparasites are widely considered to be a major constraint on their production and welfare [2]. Infection by gastrointestinal and bronchopulmonary parasites primarily results in subclinical disease, but symptoms such as anorexia, diarrhoea, weight loss, weakness, rough hair coat, coughing, and in case of hematophagous species (e.g., *Haemonchus contortus*, *Bunostomum trigonocephalum*, and *Fasciola hepatica*), anaemia and oedema may be present [3]. Endoparasites are normally associated with significant production losses, with a decrease of milk yield and quality, reduced growth rate, discarded organs at slaughter and death [3,4,5]. Beyond this, predisposition to secondary infections and additional costs for veterinary intervention and treatments should also be considered, as well as the risk of the onset of anthelmintic resistance (AR) when treatments are performed.

In Italy, as in most of the world, the control of endoparasites relies almost exclusively on anthelmintics [4,6,7], which are commonly administered without any diagnostic support [8]. Moreover, the relative dominance of sheep production in developed countries determined a lack of caprine-oriented studies and the knowledge about goats has been, for a long time, directly inferred from sheep [9]. However, compared to sheep, goats develop higher parasite burdens and their metabolism requires higher dosages of xenobiotics for appropriate efficacy, and thus resistance selection due to underdosage might have been facilitated in goats [2,7,9,10]. The frequent and often inappropriate use of drugs has led to a global spread of AR and early reports have been published in Italy in both sheep [11,12] and goats [13,14]. In order to safeguard the future of ruminant production and to ensure its long-term sustainability, there is a clear need to rethink parasite control practices [15]. Currently, strategies more widely agreed on by the scientific community to slow the development of resistance include the use of integrated control schemes, based on alternative measures to substitute or complement the action of anthelmintic drugs [16,17], and the refugia-based approach to properly target anthelmintic treatments [15,18]. In this context, regular monitoring remains central to avoid unnecessary drenching and to detect early changes in anthelmintic susceptibility. In the past years, several patho-physiological indicators have been proposed as potential markers to identify animals requiring treatment, such as the FAMACHA assessment, which estimates the degree of anaemia caused by blood-feeding parasites [18]. However, previous experiences in Italy showed low sensitivity in detecting anaemic sheep [19]. Besides, non-hematophagous species cannot be monitored with this system. Therefore, despite its inability to specifically identify *Haemonchus,* unless more expensive procedures (i.e., coproculture, PNA binding or molecular tools) are also included, faecal egg count (FEC) remains, at present, the most common diagnostic tool in small ruminant practice [20]. The interpretation of FEC results need to remain flexible, since operational (i.e., storage conditions and laboratory technique), physical (i.e., fecal dry matter and feed intake) and biological (i.e., seasonality and parasite prolificity) factors may influence the outcome. For this reason, the wide use of an EPG (eggs per gram of faeces) threshold (commonly around 500 EPG) for the application of the treatment is incorrect. Indeed, this proves even more true when pooled samples are used [21]. Parasites have a highly aggregated distribution, meaning that about 80% of the worms are found in just 20% of the hosts, whereas the remaining host population usually shows lower burdens [22]. Therefore, information collected by individual analysis is more complete and better portrays the farm infection status compared to pooled samples. However, the major cost of individual FECs remains its main limitation and requires balancing the sample size in order to obtain an accurate idea of the actual burden of the flock while limiting the costs. Regarding this, Gregory and Woolhouse [23] concluded that small samples from an aggregate population have a low probability of detecting few hosts with higher burdens, thus likely leading to mean abundance underestimation. The more the sample size decreases and the aggregation increases, the higher is the chance of underestimation of the true value. Few studies have investigated these aspects from a practical point of view, generally focusing on the use of composite samples [24], and providing only generic indications for sample size determination (e.g., “10 animals within a group”) [21]. Above all, host population size has barely been considered, and therefore sample size determination is usually not properly correlated to the farm size.

The main objective of this study was to investigate the epidemiological features of endoparasite infection in dairy goats in the lowlands of north-eastern Italy, which was through a cross-sectional survey based on a new formula developed for a farm-tailored sample size determination. In the study area, which has been poorly investigated till now, goats have a minor economic relevance compared to other livestock and data about their endoparasites are only available for the mountainous part of north-eastern Italy [4] or for bordering regions [7,14,25,26].

## 2. Materials and Methods

This study was performed involving 20 dairy goat farms located in Veneto and Friuli-Venezia Giulia, two regions of north-eastern Italy. The farms were selected either in collaboration with local veterinary practitioners or by contacting a regional farmer’s association. Farmers willing to participate were asked to answer a questionnaire, part of which was submitted to the farm veterinarian, if present. Subsequently, individual fecal samples were collected from 13 of the above-mentioned farms for the parasitological analysis.

### 2.1. Questionnaire Survey

The questionnaire was submitted to the farmers in order to collect data about farm structure and management, including herd size, breeds, other livestock in the farm, general husbandry practices and feed management, including pasture, if present. Farmers were also asked about their perception of problems and control measures related to parasites. Farm veterinarians were asked about monitoring and drenching practices for endoparasites, clinical and copromicroscopical monitoring, timing and frequency of treatment, dosages, choice and rotation of anthelmintic, management of clinical cases and of newly introduced animals. If no veterinarian in the farm was involved in endoparasite control, this part of the questionnaire was also submitted to the farmer (7/20 farms).

### 2.2. Theoretical Basis for Sample Size Determination

It is known that negative binomial is the better model for parasites in host populations [22]. Negative binomial distribution is characterized by two parameters: population mean (*µ*) and *k*. Dealing with parasites, the mean corresponds to abundance, while *k* is negatively related to parasite aggregation (lower *k* implies higher aggregation). These two parameters are linked by the following relationship:$$k=\frac{\mu^{2}}{var-\mu }$$
where *var* is the variance of the distribution.

The population variance is, therefore:$$var=\frac{\mu^{2}}{k}+\mu$$

In order to accurately estimate the mean of FECs on individual farms, both the aggregation parameter *k* and the population (farm) size *N* have been taken into account for sample size (*n*) determination.

The standard error of mean (*SE*) for infinite populations is equal to the square root of *var/n*, but should be corrected for finite populations as follow:$$SE=\sqrt{\frac{var}{n}\times \frac{N-n}{N-1}}$$

Therefore, for a negative binomial distribution
$$SE=\sqrt{\frac{\frac{\mu^{2}}{k}+\mu }{n}\times \frac{N-n}{N-1}}$$

Then, the 95% confidence interval (CI) of the sample mean (*m*) can be pragmatically approximated using the t-Student distribution with 9 degrees of freedom (that is conservative for n ≥ 10) as follow: 95%CI=m±2.26×SE.

Or, for *n* ≥ 30 using the Normal approximation, 95%CI=m±1.96×SE.

The desired sample size should therefore be calculated, given the population size *N*, the expected mean µ and the desired precision (*p* = 2.26 × *SE* or *p* = 1.96 × *SE*) as follows:$$n=\frac{Nm^{2}+Nmk}{k\times SE^{2}\times \;\left(N-1\right)\;+\;m^{2}+mk}$$

This was after estimating a reasonable *k* parameter from empirical data.

### 2.3. Sampling Approach

Individual fecal samples were collected from the rectum of 264 goats from 13 farms rearing Camosciata, Saanen or Murciana breeds. Sampled animals were randomly selected among clinically healthy goats. The sampling was performed, with care of avoiding parturition period, between November 2018 and September 2019, no less than 6 months after the last anthelmintic treatment and according to the farm-tailored approach previously described. The minimum sample size was determined according to the formula explained in Section 2.2, considering a *k* = 0.92 according to the maximum likelihood estimation performed using strongyles FECs data observed in a preliminary sample of 47 adult goats, and a precision for the estimation of the mean output of the flock equal to 50% of the expected mean output of the sampled animals. In addition, whenever possible, a few extra samples (2–7 per farm) were collected from females, males and young animals (<1 year of age). The samples were transported and stored at about +4 °C until laboratory analyses, which were carried out within 2–3 days.

### 2.4. Laboratory Analysis

All individual samples were analyzed quantitatively to estimate the fecal eggs output through FEC. Additionally, an equal amount of feces from each animal was used to prepare a pooled sample to be analyzed for first stage larvae of lungworm and to perform a coproculture whenever possible.

Individual FECs were performed by a modified McMaster method [27], which is commonly used at the Parasitology laboratory of the Department of Animal Medicine, Production and Health at the University of Padova. For each individual, 5 g of feces were used, if necessary reduced to 2 g when the amount of feces collected was not enough. Nematodes were identified as *Skrjabinema*, *Trichuris*, *Capillaria*, *Nematodirus*/*Marshallagia*, *Strongyloides* or gastrointestinal strongyles (when genus was not morphologically distinguishable). The number of eggs/oocysts per gram (EPG/OPG) was calculated for cestode and nematode in all samples, while *Eimeria* spp. infection was only evaluated quantitatively in young animals (<1 year of age). When all samples from a single farm were negative to helminths, we only analyzed a number of samples sufficient for demonstrating that prevalence on that farm was under 20% for all helminthic groups, considering a 90% CI and the total population equal to the farm size [28].

Subsequently, 5–8 individual samples were grouped according to the farm management and housing partition of the flock and pooled samples were composed with 5 g of feces from each animal and thoroughly mixed up. Pooled samples were processed by modified Baermann technique to detect first-stage larvae (L1) of lungworms. For each analysis, 5 g of pooled faeces were used. Different genera of lungworms were identified according to morphological keys [29]. When mean abundance of gastrointestinal strongyles (GIS) in respective individual samples was above 400 EPG, the remaining amount of pooled faeces was also cultured for 7 days at about +26 °C to identify the genus of third-stage larvae (L3) of gastrointestinal nematodes, which were recovered from coprocultures by the Baermann technique. The first 100 randomly selected larvae were identified to the generic level as *Trichostrongylus*/*Ostertagia*/*Teladorsagia*, *Haemonchus*, *Oesophagostomum*/*Chabertia* and *Bunostomum*/*Gaigeria*, according to morphological keys [27,29], as per the diagnostic identification procedure at the Laboratory of Parasitology of the University of Padova. When fewer than 100 larvae were isolated from a pool, all larvae were identified.

### 2.5. Data Analysis

Description of general characteristics of investigated farms, prevalence of infection and abundance (mean number of OPG/EPG) were displayed using simple descriptive statistics that considered all the analyzed samples. Confidence interval (*CI*) and standard error (*SE*), respectively, of the prevalence and the abundance of the overall population, were calculated considering an infinite population.

The influence of different factors (use of pasture, sex and age class) on the parasitic burden of different groups was investigated using multiple approaches (negative binomial regression); robust standard errors clustered for the farm were computed, in order to adjust for possible similarity among individuals belonging to the same farm [30]. Firstly, the analysis was performed on all sampled animals, considering three age classes (<1 year, 1–2 years, >2 years), and secondly only for adult females to investigate interaction between the use of pasture and age classes (1–2 years, >2 years). Statistical analyses were performed using the software STATA^®^ 12.1.

Within individual farms, epidemiological indexes were based on a number of samples equal to the minimum sample size, randomly excluding the few exceeding samples. The confidence interval of the abundance estimated for each farm was calculated according to the following formula:$$SE=\sqrt{\frac{\sum_{i=1}^{n}{\left(m-\mathrm{EPG}\left(i\right)\right)}^{2}}{n-1}\times \frac{N-n}{n\left(N-1\right)}}$$

We then calculated that:
m±2.26×SE=95%CI when *n* < 30 (conservative approximation using Student’s t-distribution with 9 degrees of freedom).m±1.96×SE=95%CI when *n* > 29 (approximation using z = standard normal distribution).

Where *SE* is the standard error, *m* is the mean abundance, EPG(i) is the EPG emitted by animal i, *n* is the sample size and *N* is farm size.

## 3. Results

### 3.1. Farm Questionnaire

The 20 dairy goat farms surveyed in this study were distributed over the whole region of Veneto (except for the montainous province of Belluno) and in the provinces of Udine and Pordenone, in Friuli-Venezia Giulia Region (Figure 1). Farms were located between 2 m a.s.l. and 760 m a.s.l. of altitude, but most of them were in the lowlands, under 200 m a.s.l. (17/20). The mean size of the flock was 85.7 animals, ranging between 37 and 208 adult goats. On average, 4.6% of the total adults were bucks. The dominant breed was Camosciata (present in 16/20 farms), followed by Saanen (7/20), both cosmopolitan breeds highly selected for milk production. The only other breed reared in the present study was Murciana (2/20). Other livestock species were present in over half of the farms (11/20). Goats from seven flocks were reared exclusively indoor, while the remaining had access to an external paddock or a pasture. Access to the outdoor area was either continuous all year round or limited to the warm season, depending on the farm. A single flock implemented a pasture rotation. Management of kids (younger than one year of age) and bucks differed from the rest of the flock as they were usually kept in separate pens and had no outdoor access.

Only 10% (2/20) of the farmers performed regular monitoring of endoparasites, while in the remaining farms, the use of coprological analysis was occasional or limited to the diagnosis of suspected clinical cases. A routine treatment of the adults against helminths was performed by 70% (14/20) of the farmers, using drugs belonging to benzimidazoles and macrocyclic lactones classes. Annual treatments were usually performed between November and February (taking advantage that milking goats are in their dry period). On a few farms (3/14), a regular additional treatment was administered during spring or summer.

### 3.2. Epidemiological Indexes

Considering all sampled individuals, 93.2% were positive for parasites. The values of prevalence and abundance of each parasite group are reported in Table 1. Coccidia were the most common group (78.4%), followed by gastrointestinal strongyles (GIS) (37.9%), *Strongyloides* (28.4%), *Skrjabinema* (18.9%), *Trichuris* (8.0%) and *Nematodirus*/*Marshallagia* (0.4%). Neither *Capillaria* nor cestoda were found. Mean abundance was high for coccidia (43,003.5 OPG in animals < 1 year of age) and GIS (484.1 EPG), while the other parasites generally showed very low output levels. Coproculture was carried out on eight pools and a total of 614 larvae were identified. *Haemonchus* resulted as the dominant genus (71%), but *Trichostrongylus/Ostertagia/Teladorsagia* (20%), *Oesophagostomum/Chabertia* (9%) and *Bunostomum/Gaigeria* (< 1%) morphotypes were also present. Identification of lungworm first-stage larvae following the Baermann technique indicated the sole presence of *Muellerius* in five pools (over a total of 29 analysed).

### 3.3. Factors Influencing Parasite Burden

The influence of pasture and age on the presence and abundance of gastrointestinal parasites was displayed through a descriptive statistic and analyzed by means of multivariable models. The influence of sex was also evaluated, but results need to be interpreted in the light of the limited size of the male sample and of the specific husbandry conditions of bucks. Table 2 shows the differences in prevalence and egg output among groups identified by the considered variables. Coccidia were similarly present in all groups, except for males and young goats, in both of which higher prevalences (≥90%) were reached.

The negative binomial regression model was conducted, including 256 animals (for eight kids it was not possible to identify the sex) (Table 3) and the results showed that the use of pasture was associated with a higher presence of GIS (*p* < 0.001) and with a lower presence of *Strongyloides* (*p* < 0.001). Regarding the influence of sex, males had a lower abundance of GIS (*p* < 0.01) and *Skrjabinema* (*p* < 0.05) compared to females. Finally, in relation to age, adults were significantly more exposed to *Strongyloides* infection (*p* < 0.001) than young animals, while no such significant difference emerged for GIS (*p* > 0.05). Significant differences for *Skrjabinema* only emerged between young animals and adults 1–2 years old, with the latter being more affected (*p* < 0.01). The second negative binomial regression model (Table 4) focused more in depth on how the two factors ‘use of pasture’ and ‘age class’ interacted, considering only adult females in the analysis (n = 231). The interaction was significant for GIS (coeff = 1.66, *p* < 0.05) and *Strongyloides* (coeff = −1.94, *p* < 0.05), but not for *Skrjabinema* (*p* = 0.868). In particular, the use of pasture has a stronger effect on younger adults for GIS compared to older goats, while the protective effect of pasture for *Strongyloides* infection is more evident in older animals. Figure 2 provides a graphical explanation on the way these two risk factors interact for GIS (Figure 2a), *Strongyloides* (Figure 2b) and *Skrjabinema* (Figure 2c) infection.

### 3.4. Farm-Tailored Monitoring

The results from farm monitoring are summarized in Table 5. In this analysis, only the minimum sample size for each farm was considered (n = 214), randomly excluding the few exceeding samples. Bucks and kids were excluded due to their different husbandry practices. Coccidia were ubiquitous, with prevalences higher than 75% in 10/13 flocks, while *Skrjabinema* and *Trichuris* outputs and prevalence remained extremely limited in all the farms, in agreement with the above-mentioned epidemiological indexes. *Strongyloides* was able to occasionally exceed an emission level of 50 EPG and in three flocks it was recovered in over half of the sampled animals. The trend was in many cases antithetical in terms of both prevalence and abundance to that of GIS, whose presence has the greatest relevance from a practical standpoint. When present (8/13 farms), GIS showed very distinct prevalence values, ranging from 20% to 100%. The same heterogeneity was displayed for emission levels, which ranged from a minimum abundance of 12.9 EPG up to 2033.3 EPG in positive farms. Abundance of GIS and relative 95% *CI* are graphically presented in Figure 3 in relation to the main characteristics of the farms. The confidence interval provided information about the reliability of the abundance estimation. In farm VG1, the wide range of the 95%*CI* was the result of the exceptional aggregation of parasites in the sample, since 11/17 goats were negative, 3/17 had a very low output (<120 EPG), but the remaining had high or very high emission levels (940–6060–11,280 EPG).

## 4. Discussion

The questionnaire survey provided an overview of the structure of goat farming in the study area, with a focus on endoparasite control practices. The farming system was based on intensive/semi-intensive medium to large farms with owned land area. Milk was mostly transformed in-farm and cheese directly sold to consumers [31]. Anthelmintic treatments were less frequent compared to other European countries [32,33,34], confirming what has already been reported in Northern Italy by Lambertz et al. [4], Manfredi et al. [7] and Zanzani et al. [14]. The lower reliance on drugs for parasite control is likely among the key factors accounting for the low incidence of AR in Italy [6]. Nevertheless, the onset of resistance is a biological consequence of the use of drugs and the information collected in this study underline the need to improve parasite management on goat farms of north-eastern Italy. Indeed, 70% of farmers performed at least one anthelmintic treatment per year, but drenching practices were often poor in terms of anthelmintic dosage choice and pharmaceutical class rotation. Additionally, in agreement with previous studies performed in Northern Italy [4,7,14], we recorded scarce use of coprological analysis, which are routinely implemented in just one tenth of farms.

Despite the general underestimation of the problem, endoparasites were identified in 93.2% of animals. Coccidia were ubiquitous, being the only group present in all farms, with a prevalence of 78.4%, which is in line with other data collected in Italy [35]. Their abundance was only calculated for animals <1 year of age and just one of them tested negative. Oocyst count provided a wide range of values, with a mean of 43,003.5 OPG. It should be noted that all sampled animals did not show symptoms; therefore, this value can be considered normal in healthy animals, although it seems very high. In fact, healthy individuals may pass over a million OPG, whereas others may die of coccidiosis with less than 10,000 OPG of faeces [36]. In this case, good hygienic conditions and absence of overcrowding and stress may have contributed to the healthy status of the animals. *Capillaria* and cestoda were absent in all samples. While this may be due to the limits of sensitivity of the McMaster technique, it still points out that, if present, these parasites have very low prevalences and abundances and therefore they do not represent an issue for goat production and health in the study area. This is likely also true for *Trichuris*, *Skrjabinema* and *Strongyloides*, whose abundances were minimal, although *Strongyloides* reached a significant output (1140 EPG) in one individual. About one-third (37.9%) of sampled goats were positive to GIS, similar to data recorded by Di Cerbo et al. [24] and Manfredi et al. [7] in other goat farms of Northern Italy. The relative dominance of *Haemonchus* in our study was unprecedented in Northern Italy, even if its increased presence in the area was already observed in recent years [14]. In temperate countries, the genera *Trichostrongylus*, *Ostertagia* and *Teladorsagia* were historically the most represented, while *Haemonchus* was more typical of tropical and sub-tropical regions [16]. Our finding, which is particularly alarming given the high pathogenicity of *Haemonchus*, is in agreement with recent studies [37], suggesting that parasite epidemiology is changing in temperate countries, such as Italy, arguably in relation to climate change. Further investigations will be required in the future to deal with the evolving infection patterns. Above all, the egg output for a few of the pastured herds should be monitored during a year to determine exactly when the high-risk period for goat deaths is occurring. In areas where Haemonchus is known to be the dominant parasite, an evaluation of anaemia through FAMACHA should be integrated to support treatment decisions. FAMACHA estimates the degree of anaemia by assessing the color of the conjunctival mucosal membrane on a five-point scale. This system has been validated in goats [38,39] and has proved to be a practical and low cost tool to target treatments more effectively [18]. However, it is suitable for the identification of blood-feeding parasites (e.g., *Haemonchus contortus*) only, while parasite infections tend to be of mixed-origin [15].

GIS prevalence and abundance showed marked differences between farms, corroborating the influence of management practices for the presence of parasites. In our study, the most significant (*p* < 0.001) risk factor for strongyle infection was the use of pasture and very low to null emission levels were recorded on farms devoid of a grazing area. It is well-known that GIS infection is typically associated with pasture, as free-living stages find there a suitable environment for their growth [7,26], but our results underline once again the need to improve grazing management in order to reduce the reliance on pharmacological treatments and to achieve an integrated and more sustainable control of endoparasites. However, indoor rearing exposed goats to a significantly (*p* < 0.001) heavier *Strongyloides* infection, both in terms of prevalence and abundance. According to Manfredi et al. [7], flocks are more at risk in the case of high stock densities and scarce sanitary management. When looking at the descriptive statistics, males reached a higher prevalence of coccidia and higher prevalence and abundance of *Strongyloides* compared to females, while GIS and *Skrjabinema* seem to be more present in the latter. However, the reliability of our results is certainly limited by the small size of the male sample and by their specific management, as previously underlined. The multivariable model for nematodes highlighted no significant effect of sex on Strongyloides abundance, probably because of the inclusion of the presence/absence of pasture in the model. Literature is rather contrasting about the influence of sex on parasitological indexes, with females [40,41] and males [42,43] described alternatively as more predisposed. In other studies [44,45], no significant differences were recorded. Coccidia were widely present (95.7%) in goats less than one year old, with prevalence decreasing with age, likely for the development of a strong immunity [4,36,43]. On the contrary, considering the remaining parasites, young animals showed the lowest values in terms of both prevalence and abundance. The increasing levels of helminth infection with age have been associated with prolonged exposure to third-stage larvae [46] and with the low ability of goats to develop an immune response against gastrointestinal nematodes [47]. Nevertheless, some authors [4,41,48] observed no significant difference among animals of different ages, while others [43,44,45,47] found higher values in younger animals, arguing that with age and exposure, goats can actually develop a certain immunity. However, the same husbandry conditions described for males usually apply to young animals (<1 year old) too. Concerning the two risk factors ‘use of pasture’ and ‘age class’ and their interaction in adult females, 1–2 year old goats and grazing animals had significantly higher emissions of GIS than older animals (*p* < 0.001) and non-grazing animals (*p* < 0.05) respectively. The interaction revealed a significantly higher (*p* < 0.05) increase in EPG in 1–2 year old goats compared to the older counterpart when pasture was available, probably due to the lack of previous exposure to pasture parasites in younger adults [43,44,45,47]. Since GIS represent the main parasitic threat to goat production and health, this information can serve as a possible parameter to selectively target treatments on part of the flock.

The scientific community identifies in targeted selective treatments (TST) the ideal drenching strategy to tackle AR, but in common practice its implementation holds some major limits, mainly due to the lack of widely available user-friendly decision support systems [15]. Accordingly, while TST should be promoted as the only responsible drenching strategy, whole-flock targeted treatments (TT) are likely to remain for the next future the main control strategy for endoparasites. This is currently considered acceptable in areas where AR is rare, such as Italy, but farmers using TT should monitor their flock to detect early changes in susceptibility and, should that happen, turn immediately to a TST approach [15]. Despite being criticized for its poor correlation with parasite burden and animal performance [21] and for its lack of standardization [49], FEC represents a key indicator to target whole-flock treatments effectively [18]. Over-reliance on strict thresholds for treatment can be misleading, but when a careful evaluation of individual features (i.e., clinical symptoms, decreased production, reproductive status), farm management (i.e., use of pasture) and epidemiological data (i.e., local epidemiological patterns, farm history) precedes the interpretation of egg counts, FEC proves to be a useful tool, especially if regular monitoring is performed [21]. However, parasite aggregated distribution adds further intrinsic difficulties. The concentration of most parasites in the minority of the host population complicates the detection of the real infection status, especially when small samples are involved, and exposes to the risk of underestimation [22]. For the same reason, individual analyses are deemed to be more informative than counts performed on pooled samples [21], but the consequent higher cost requires to keep the number of sampled individuals as low as possible. Our study attempts to answer the need for clearer indications on how to monitor endoparasites, since our literature research underlined a major lack concerning this issue. With the purpose of providing farmers and veterinary practitioners a useful and reasonably applicable tool for common practice, we propose a new monitoring approach based on individual FECs, where sample size (calculated as in Section 2.2) is tailored on the host population (farm) size. Using the formula provided in Section 2.5, the calculated mean is associated to a 95% CI, which reflects the heterogeneity of the individual FECs, and allow for an assessment of the reliability of the estimation. *Strongyloides*, *Skrjabinema* and *Trichuris* usually show very low outputs and thus they are unlikely to significantly affect the total emission levels. Hence, the discussion focuses exclusively on GIS.

Mean abundances found in the investigated farms can be clustered in 3 range values: null/low (<300 EPG), intermediate (300–1000 EPG) and high/very high (>1000 EPG) emission levels. Our survey suggests that the estimation of GIS burden is highly reliable when a moderately ample 95% CI is associated with mean abundances that are high/very high (farms FR1, FR2) or low (farms FR6, VA1). In these cases, the results of the farm surveys provided clear indications to decide about the need or not for an anthelmintic treatment, respectively. Farm VG1 represents an interesting exception. Its mean abundance (1084.7 EPG) is completely diverted by three outliers, suggesting a high infection burden, despite the negative (11/17) or very low (3/17) FECs of most sampled goats. In this specific case, the extremely ample confidence interval (0–2473.5 EPG) proved an unusually high level of aggregation of the parasite population and suggested the need for further in-depth analysis. Lastly, intermediate emission levels (farm FR3, VA2, and VA3) provided uncertain results, since their 95% CI limits encompassed the EPG threshold used in common practice (500 EPG) for deciding when an anthelmintic treatment would be recommended. In these cases, as previously mentioned, it is important to consider individual, management and epidemiological factors. Above all, regular (e.g., once/twice a year) monitoring may allow for a sounder decision, based also on the variation of the output from the previous sampling. Evidently, parasitological exams should be scheduled prior to treatment.

## 5. Conclusions

Veterinary practitioners occupy the front line in the fight against AR and the proposed farm-tailored monitoring approach could represent a cost-effective decisional tool to target treatments more effectively. However, farmers tend to be reluctant to pay for coprological testing, as parasite infections are often subclinical and the long-term risks of drug resistance are not fully understood. For this reason, while frequent monitoring provides the best results, it would not be a realistic option. Through data obtained on the analysis of risk factors, we suggest in farms where animals have access to a grazing area to monitor for parasites at least twice a year. Particular attention should be addressed to sampling young adults (1–2 years), given their higher susceptibility to pasture-related parasites compared to older animals. This information could also be useful to target treatments on a specific group of animals only (i.e., young adults), according to the principles of refugia-based strategies. Where animals are kept only indoors and GIS are absent or present with very low burden, monitoring could be implemented at least once every two years in order to verify the steadiness of the infection level.

The relative dominance of *Haemonchus*, a strongyle species of primary concern, is a major finding of this study, as it can quickly kill heavily infected animals without noticeable symptoms. When *Haemonchus* is known to be the dominant parasite, a FAMACHA assessment should support the interpretation of FEC results, possibly leading to a TST drenching approach. FEC, however, provides additional useful information about parasite burden and distribution in the host population and is essential to monitor non-hematophagous species, such as *Strongyloides*, coccidia and other trichostrongylids, which are known to be present in Northern Italy [4,14].

Finally, the absence of cestoda and *Capillaria* and the scarce presence of *Trichuris* and *Skrjabinema* suggest that these parasites are of limited relevance in the study area.

As anthelmintic resistance spreads at an alarming pace, there is an urgent need for clear and practical guidelines for endoparasites monitoring. The first study using this farm-tailored sample size led to promising results and to a fairly accurate estimation of mean abundance, but further investigations are necessary to confirm its practical value. Simultaneously, we conducted the first epidemiological survey in goat farms of north-eastern Italy. Despite the wide presence of gastrointestinal parasites, control strategies showed some major gaps. Above all, we highlight the scarce use of coprological analyses, in agreement with previous reports from Northern Italy [4,7]. The lack of diagnostic support prevents the accurate identification of flocks requiring treatment, exposing farmers to the risk of economic losses when highly parasitized animals are left untreated. Besides, the redundant use of anthelmintics leads to an increased selective pressure for resistance. Accordingly, one of the main challenges for the next future is to encourage farmers to include parasitological testing in the routine health surveillance of the flocks, since the knowledge of the infection status is essential to set up a control strategy that is sustainable in the long-term. The availability of clear and practical guidelines for monitoring parasitic burden represents the first step towards this objective.

## Figures and Tables

**Figure 1 vetsci-08-00069-f001:**
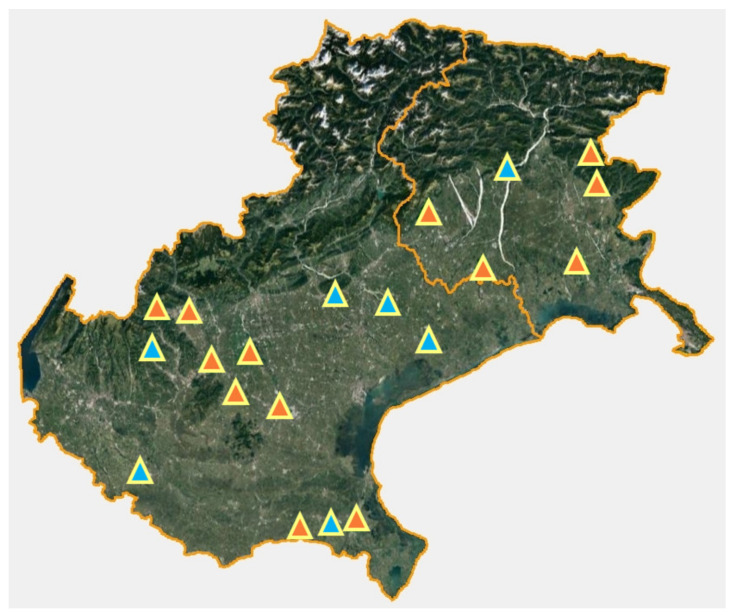
Location of the studied farms in the two regions of Veneto (on the left) and Friuli-Venezia Giulia (on the right). Farms exclusively surveyed through questionnaire (n = 7) are marked in blue, while orange designates those also sampled for coprological analysis (n = 13).

**Figure 2 vetsci-08-00069-f002:**
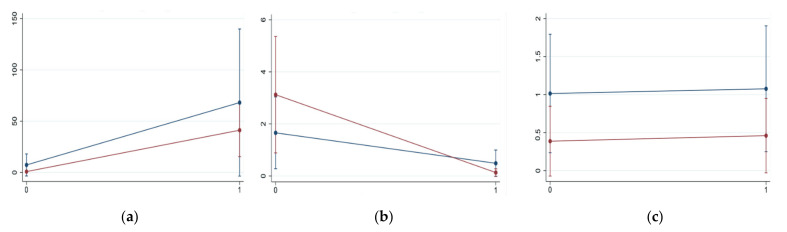
Interaction between use of pasture (0 = NO; 1 = YES) and age class (in blue = 1–2 years old; in red = older than 2 years of age) in the negative binomial regression model for GIS (**a**), *Strongyloides* (**b**) and *Skrjabinema* (**c**) conducted for adult females (n = 231). In *y*-axis the actual counts at McMaster observation (i.e., EPG/20) are reported.

**Figure 3 vetsci-08-00069-f003:**
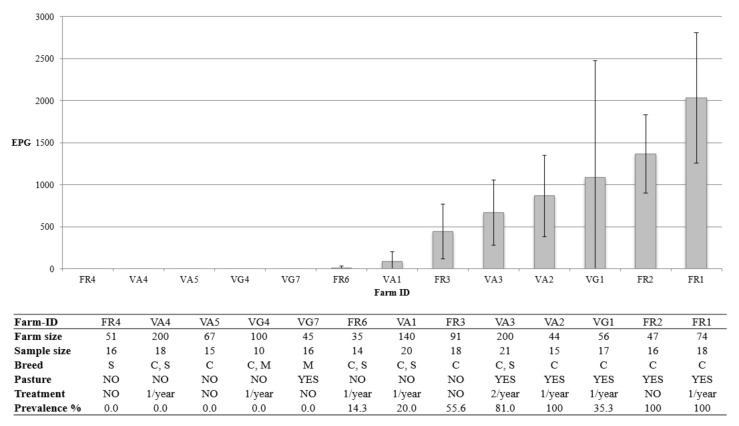
Mean abundance and 95% confidence interval of GIS in relation to the main characteristics of individual farms (n = 214). EPG = eggs per gram of faeces; C = Camosciata; M = Murciana; S = Saanen.

**Table 1 vetsci-08-00069-t001:** Epidemiological indexes of gastrointestinal parasites in the study area (n = 264).

Parasite	P %	A (±SE)	Min–Max
Coccidia	78.4	43,003.5 ± 18,327.6 *	0–305,640 *
GIS	37.9	484.1 ± 65.1	0–11,280
Strongyloides	28.4	25.3 ± 4.8	0–1140
Skrjabinema	18.9	12.5 ± 2.0	0–240
Trichuris	8.0	7.0 ± 2.1	0–420
Nematodirus/Marshallagia	0.4	0.1 ± 0.1	0–20
Capillaria	0.0	-	-
Cestoda	0.0	-	-
Total	93.2		

* data refer to animals <1 year of age (n = 23). Note: P = Prevalence; A = Abundance; SE = Standard Error; GIS = gastrointestinal strongyles.

**Table 2 vetsci-08-00069-t002:** Prevalence and abundance of coccidia, GIS, *Strongyloides* and *Skrjabinema* in different groups, according to sex, age and use of pasture.

		n	Coccidia	GIS		Strongyloides		Skrjabinema	
Risk Factor			P%	P%	A	P%	A	P%	A
			(95% CI)	(95% CI)	(SE)	(95% CI)	(SE)	(95% CI)	(SE)
Use of pasture	No	148	79.7%	14.2%	79	39.9%	41	18.2%	12
			(73.2–86.2)	(8.6–19.8)	(18)	(32.0–47.8)	(7)	(12.0–24.4)	(2)
	Yes	116	76.7%	68.1%	1001	13.8%	5	19.8%	13
			(69.0–84.4)	(59.6–76.6)	(97)	(7.5–20.1)	(1)	(12.5–27.1)	(2)
Sex *	F	246	77.2%	39.8%	517	28.0%	24	19.9%	13
			(72.0–82.4)	(33.7–45.9)	(74)	(22.4–33.6)	(5)	(14.9–24.9)	(2)
	M	10	90.0%	20.0%	52	60.0%	84	10.0%	2
			(71.4–100)	(0–44.8)	(8)	(29.6–90.4)	(9)	(0–28.6)	(0)
Age class (years old)	<1	23	95.7%	8.7%	42	0%	0	4.3%	1
			(87.4–100)	(0–20.2)	(9)	(0–0)	(0)	(0–12.6)	(0)
	1–2	112	80.4%	33.0%	567	32.1%	28	25.9%	20
			(73.0–87.8)	(24.3–41.7)	(91)	(23.5–40.7)	(4)	(17.8–34.0)	(3)
	>2	129	73.6%	47.3%	491	30.2%	27	15.5%	8
			(66.0–81.2)	(38.7–55.9)	(55)	(22.3–38.1)	(7)	(9.3–21.7)	(1)

* For 8 animals (all <1 year old) the sex was not determined. Note: P = Prevalence; A = Abundance; 95% CI = Confidence Interval at 95%; SE = Standard Error; GIS = gastrointestinal strongyles.

**Table 3 vetsci-08-00069-t003:** Results of negative binomial regression model for GIS, *Strongyloides* and *Skrjabinema*, considering all sampled animals (n = 256). Significant values are highlighted in bold.

Risk Factor		GIS		Strongyloides		Skrjabinema	
		Coef.	*p* Value	Coef.	*p* Value	Coef.	*p* Value
Use of pasture	No	ref		ref		ref	
	Yes	**3.19**	<0.001	**−2.22**	<0.001	0.11	0.852
Sex	F	ref		ref		ref	
	M	**−2.47**	0.002	0.63	0.193	**−2.12**	0.037
Age class (years old)	<1	ref		ref		ref	
	1–2	0.58	0.648	**19.26**	<0.001	**2.71**	0.009
	>2	−0.86	0.431	**19.21**	<0.001	1.79	0.129

Note: GIS = gastrointestinal strongyles; ref = reference.

**Table 4 vetsci-08-00069-t004:** Interaction between the two factors ‘use of pasture’ and ‘age class’ in negative binomial regression model for GIS, *Strongyloides* and *Skrjabinema*, considering only adult females (n = 231). Significant values are highlighted in bold.

Risk Factor		GIS		Strongyloides		Skrjabinema	
		Coef.	*p* Value	Coef.	*p* Value	Coef.	*p* Value
Use of pasture	No	ref		ref		ref	
	Yes	**2.23**	0.015	**−1.22**	0.072	0.06	0.915
Age class (years old)	1–2	ref		ref		ref	
	>2	**−2.17**	<0.001	0.63	0.289	−0.96	0.063
Interaction (age class > 2/use of pasture Yes)		**1.66**	0.026	**−1.94**	0.012	0.11	0.868

Note: GIS = gastrointestinal strongyles; ref = reference.

**Table 5 vetsci-08-00069-t005:** Prevalence and mean abundance with 95% confidence interval of gastrointestinal parasites recovered in each farm. Data refer to adult females (n = 214). * In these farms it was not possible to collect as many samples as expected.

Farm ID	N	n	Coccidia	GIS			Strongyloides			Skrjabinema			Trichuris		
			P%	P%	A	95% CI	P%	A	95% CI	P%	A	95% CI	P%	A	95% CI
FR1	74	18	94.4	100	2033.3	1257.7–2808.9	27.8	10.0	0.3–19.7	27.8	5.6	1.3–9.9	0.0	0.0	0–0
FR2	47	16	62.5	100	1368.1	904.4–1831.8	0.0	0.0	0–0	6.3	1.3	0–3.6	6.3	2.5	0–7.1
FR3	91	18	77.8	55.6	442.2	116.5–767.9	44.4	38.9	11.5–66.3	27.8	24.4	0.1–48.7	11.1	7.8	0–21.4
FR4	51	16	81.3	0.0	0.0	0–0	56.3	38.8	6.3–71.3	0.0	0.0	0–0	6.3	1.3	0–3.7
FR6	35	14	21.4	14.3	12.9	0–33.2	42.9	18.6	6.0–31.2	21.4	8.6	0–17.5	0.0	0.0	0–0
VA1	140	20	80.0	20.0	89.0	0–201.7	55.0	58.0	22.0–94.0	30.0	22.0	1.1–42.9	20.0	11.0	0–22.6
VA2	44	15	40.0	100	868.0	381.9–1354.1	6.7	1.3	0–3.8	6.7	2.7	0–7.6	13.3	17.3	0–47.0
VA3	200	21	85.7	81.0	666.2	277.6–1054.8	28.6	10.5	1.3–19.7	42.9	40.0	11.6–68.4	9.5	5.7	0–14.7
VA4 *	200	18	94.4	0.0	0.0	0–0	55.6	103.3	0–237.9	44.4	36.7	8.2–65.1	5.6	3.3	0–10.5
VA5 *	67	15	86.7	0.0	0.0	0–0	40.0	15.3	0–31.3	6.7	1.3	0–4.0	6.7	2.7	0–8.0
VG1	56	17	76.5	35.3	1084.7	0–2473.5	0.0	0.0	0–0	23.5	14.1	0.7–27.5	11.8	3.5	0–8.2
VG4	100	10	90.0	0.0	0.0	0–0	0.0	0.0	0–0	0.0	0.0	0–0	0.0	0.0	0–0
VG7	45	16	87.5	0.0	0.0	0–0	0.0	0.0	0–0	12.5	4.4	0–10.4	6.3	1.3	0–3.5
Total	1150	214	76.2	41.1	532.1	357.9–706.2	28.0	24.5	12.2–36.8	21.0	14.0	9.0–19.0	7.9	4.6	1.6–7.6

Note: N = farm size; n = sample size; P = Prevalence; A = Abundance; 95%CI = Confidence Interval at 95%; GIS = gastrointestinal strongyles.

## Data Availability

Data available on request due to privacy restrictions.

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
