# Peer review of "Survey on Endoparasites of Dairy Goats in North-Eastern Italy Using a Farm-Tailored Monitoring Approach"

_vetsci, 2021, doi:10.3390/vetsci8050069_

Round 1
Reviewer 1 Report
I read the manuscript by Maurizio et al., with great interest. It is an understudied area in veterinary parasitology and this is a nice, simple and novel approach to take to the studies. I have a few minor comments below but overall, it is a good, well written manuscript which adds to the literature on goat parasites.
Line 17- I would replace endoparasites with endoparasite
Line 36- maybe in economic, environmental and cultural perspectives may sound better?
Line 42- a few examples of parasites here may be nice- Fasciola hepatica, Haemonchus contortus, or ones which are an issue there?
Line 45- is it worth mentioning drug resistance here?
Line 51- sheep instead of sheeps
Line 56- delete also
Line 62- comma after context
Line 64-65- not sure feacal egg count needs capitals?
Line 66- delete however
Line 85- comma after estimate, and after accuracy
Line 87- comma after area
Line 95- maybe say that the study was performed using? Including suggests that there may be more?
Line 99- how were these samples chosen? Just at random? Or more strategically?
Line 110- maybe endoparasite rather than endoparasites?
Line 113- does negative binomial need capitals?
Line 126- maybe on individual farms?
Line 129- mean doesn’t need a capital
Line 134- again negative binomial doesnt need capitals
Line 138- please put the m in brackets to aid reading
Line 148- not needed to be in italics
Line 150- a bit more data here on distribution may be useful- ages, sex, breeds etc
Line 158- maybe a few extra samples? (but could you be more specific?)
Line 163- replace besides with additionally
Line 168- comma after individual
Line 175- replace to with for
Line 176- prevalence on that farm
Line 196- interval and standard error don’t need capitals
Line 197- comma after population
Line 124- region doesn’t need a capital
Line 1270 whats asl?
Line 127- were in plain doesn’t make sense and I cant offer a way to change it as I don’t know what you are wanting to say
Line 239- only 10% (2/20) of the farmers ….
Line 239- while in the remaining XX (isn’t a full sentence)
Line 244- On a few farms,
Line 245- comma after (3/14)
Figure 1, some of the symbols are not easy to see, could you use different shapes instead maybe? Or different colours?
Line 252- positive for parasites
Line 258- what other genera?
Line 270- I think comments about the males needs to be toned down, as the numbers are so small
Same for lines 284-285
Line 293- pasture has a strong effect ….
Table 3- maybe worth highlighting significant values in bold?
Line 308- doesn’t different husbandry of bucks and kids affect the previous data? If they are looked after separately maybe worth stating in the methods.
Line 312- what is a noteworthy emission level ?
Table 5- why are there two P% for coccidia?
Lije 334-336- reword this as unclear
Line 337- confirming what was already reported in …. (reword)
Line 340- whats AR?
Line 342- on goat farms rather than in
Line 350- with a prevalence of 78.4% which is in line … (reword)
Line 350-351- any idea why that animal was negative?
Line 352- were any animal showing any symptoms on the farms?
Line 356- comma after case
Line 367- comma after countries
Line 374- what difference, if any will the climate on these farms make? Would that differ?
Line 377- on farms rather than in
Line 385-387- difficult to say this based on small numbers of males,. Maybe just tone it down
Line 392- replace anyway with however
Line 403- ages
Line 404- comma after exposure
Line 404- replace anyway with however
Line 407- replace years with year
Line 409- replace years with year
Line 414-416- reword as unclear
Line 431-435- reword as unclear
Line 443-446- were these animals bought on the farm, or were they imports? Anything special about them?
Line 447-449- reword as unclear
Line 452- whats TST? (sorry I may have missed it)
Line 458- maybe worth here saying that it could be worth testing animals prior to treating?
Line 472- on instead of in
Line 473-474- reword as unclear
Line 489-492- reword as unclear
Reviewer 2 Report
Manuscript ID: vetsci-1159312
Title: Developing a Farm-Tailored Monitoring Approach to Assess Endoparasite Burden and Promote a Rational Use of Anthelmintics in Dairy Goat Farms
Authors: Anna Maurizio, Laura Stancampiano, Cinzia Tessarin, Alice Pertile, Giulia Pedrini, Ceren Asti, Waktole Terfa, Antonio Frangipane di Regalbono and Rudi Cassini
General Comments:
This is a very well written and well organized manuscript, however, it is unnecessarily long. The study consists of a standard producer/veterinarian questionnaire coupled with a parasitological survey involving goat farms from north-eastern Italy. The questionnaire focused on structural and management factors (including anthelmintic usage) that might influence parasite intensities within each herd. The parasitological survey consisted of helminth egg and coccidian oocyst counts from 264 animals from 13 farms collected between November 2018 and September 2019. The minimum sample size for each farm was determined based upon the population size, the expected mean and the desired precision using a formula derived from the formula for calculating levels of aggregation (k) in populations exhibiting negative binomial distributions. In these calculations of minimum sample size, they assumed that the k value for all of the farms was 0.92 based upon analysis of strongyles FECs data observed in a preliminary sample of 47 adult goats. Coccidia oocysts were the most common parasite identified followed by strongyle (i.e. trichostrongyle) eggs. The authors then evaluated correlations between results from the 2 surveys. The strongest correlation showed that goats that grazed on pastures had a significantly higher strongyle prevalence and intensity, which is to be expected. A relationship between indoor rearing and Strongyloides and coccidian intensities were also noted. Coproculture results showed that Haemonchus was the dominant strongyle egg among the 13 farms included in the survey. Based upon recent global studies, this is not surprising, but apparently, Haemonchus has not been commonly reported from this region. This is probably the most important observation from the study because it significantly influences management strategies within herds that graze on pastures. In my opinion, the largest weakness of this manuscript is the total absence of any mention of the value of FAMACHA testing among pastured goats that are likely infected with Haemonchus. Instead, the authors promote monitoring fecal egg counts (FECs) as a cost-effective decisional tool to target treatments more effectively. While epidemiological studies using FECs is extremely useful for understanding management and treatment options to limit losses caused by parasites, they are not effective at identifying risk levels for Haemonchus unless coproculture, PNA binding or molecular tools are also included to identify Haemonchus intensities. These tools are not cost-effective. With the identification of Haemonchus as the predominant trichostrongyle, this manuscript should at least explain the importance of FAMACHA as a very cost-effective tool for identifying farms and individual goats suffering from haemonchosis. There are very few livestock parasitologists still recommending whole-herd treatments, especially for small ruminants. The authors should be discouraging the use of anthelmintics other than times when they are necessary for individual animals. With regular FAMACHA testing, they can determine exactly when those times are for their production system. I also question the importance of the sampling size portion of the manuscript. This might be a topic that could be justified as a technique publication, but there is no data presented in this manuscript to justify it. There is also no attempt to discuss this approach in the context of other studies that have also considered the appropriate sample size based upon the analysis of experimental data. If this approach has been justified by other studies, then they should be simply cited by the current authors with no need to go into additional details. This manuscript is not really a study evaluating different farm-tailored monitoring approaches for assessing endoparasite burdens. The only data presented consists of survey data based upon a single approach using different sample sizes. The effects of using these different sample sizes were not evaluated and a cost-benefit analysis was not performed. The introduction and discussion of this manuscript should be condensed to cover the data that was collected. This manuscript could include a short promotion rational uses for anthelmintics in dairy goat farms from this area, but it should focus on the findings of Haemonchus in these herds. While this manuscript does provide useful regional information, the finding are not at all surprising and is of little global importance. It would probably work best as a short communication.
Title:
- Since this is really a survey study, the title should be altered to better describe the contents of the study.
Abstract:
- The abstract should also be altered to include only the information derived from the data collected in the study.
Introduction:
- The final paragraph in the introduction must be altered to describe a purpose for the study that is consistent with the data that was collected. There was no attempt to determine the actual accuracy of the data coming from the different farms while using differing sample sizes, this can not be the actual purpose for the study.
- The introduction should be altered to be consistent with the actual purpose.
- This section should briefly compare the value of evaluating herds via FAMACHA versus egg/oocysts counts in order to explain the value of each.
Methods:
- The methods are generally described in sufficient detail.
- The sample size for each farm seems to be within the range used for other surveys, and so it is not necessary to justify the sample sized using the derived formula. Instead, they could use previous recommendations from studies that provided recommendations. If the authors insist on including their sample size justifications, then they should still relate their sample sizes to the published recommendations.
- The biggest problem with the method section is the lack of specific information about when the sample were collected and when the new kids were born. The relationship between parturition and sample collection can be a significant confounder in their analysis, and the therefore, this information is needed.
- As a minor issue, the number of samples contributing to the pooled samples should be specified.
Results:
- The data and interpretations provided in the results section are generally good and the statistical analyses are very thorough.
Discussion:
- The discussion of the collected data and their relation to the published literature is well-written and appropriate.
- Lines 367-373: The authors seem to be unaware of the large number of recent studies showing Haemonchus as the predominant trichostrongyle in sheep from temperate areas. Based upon the global trend, it is not surprising that they found it to be true in their area of Italy. It would have been more surprising if it wasn’t. It is possible that climate change is affect its distribution, but there is good evidence that the effectiveness of hypobiosis allows it to survive well during environmental conditions that are lethal to the free-living pasture stage. In temperate regions, this means that the highest risk for goat deaths is after the hypobiosis period when large numbers of larvae come out of the tissues and become adults. This is why the time periods when collections were made is so important to report. They should suggest that in future studies the egg output for a few of the pastured herds should be monitored during a year to determine exactly when the high-risk period is occurring. It is likely after parturition.
- Lines 431-465: The discussion of the monitoring tool is the weakest part of the manuscript. The discussion of the exception caused by VG1 actually illustrates the weakness of the approach, where the aggregation is particularly different from the assumed value, the tool falls apart. There is good data showing that while aggregation is a very common reality in most helminth hosts, the level of aggregation varies considerably because of numerous factors, but particularly the immune status of the hosts. I also believe that the whole premise of producers using FECs to determine when and who to deworm is not valid, particularly when Haemonchus is the predominant helminth. The growing importance of FAMACHA must be incorporated into the discussion. Given the universal risk for anthelmintic resistance, I don’t know of any livestock parasitologists that are currently recommending whole herd deworming of small ruminants. There are sentences in this section that implies that the monitoring tool could be used to determine when to deworm the whole herd, and while this might be still occurring in this region, it should be discouraged in all publications.
- Lines 466-476: This final paragraph should be incorporated into the conclusion section.
Conclusion:
- This is a good place to summarize the recommendations. A major portion of these recommendations must focus on minimizing the risk from Haemonchus, which can kill goats quickly without them showing noticeable symptom. If producers are effectively controlling this species, they will also be controlling problems from the other less predominant and less dangerous trichostrongyles. This can realistically be done using FAMACHA as a monitoring tool that is very done extremely easily with dairy goats that are handled frequently.
- Monitoring for problems caused by coccidia and Strongyloides may require more traditional egg/oocyst monitoring. Recommendations for monitoring these groups should be described separately from Haemonchus. However, these parasitic problems generally cause noticeable clinical symptoms before they cause significant medical problems. They are more predictable and also more easily controlled since they are generally a problem caused by the accumulation of manure in the pens.
- A valuable part of this study is demonstrating which parasites producers from this area don’t need to worry about. If would be good to summarize this in the conclusion. If there are situations where this might not be true for some producers, then these exceptions should also be noted.
Round 2
Reviewer 2 Report
The changes made in the revised version have solved the major concerns that I had with this manuscript. The authors do illustrate the need for a more systematic approach for determined sample sizes needed for aggregated samples. Without this, their approach for sample size is valid, but not a critical issue for this relatively small regional study.